# Changes in Eating Behaviors Following Taste Education Intervention: Focusing on Children with and without Neurodevelopmental Disorders and Their Families: A Randomized Controlled Trial

**DOI:** 10.3390/nu14194000

**Published:** 2022-09-27

**Authors:** Sigrun Thorsteinsdottir, Urdur Njardvik, Ragnar Bjarnason, Anna S. Olafsdottir

**Affiliations:** 1Faculty of Health Promotion, Sport and Leisure Studies, School of Education, University of Iceland, Stakkahlid, 105 Reykjavik, Iceland; 2Faculty of Psychology, School of Health Sciences, University of Iceland, Saemundargata 12, 102 Reykjavik, Iceland; 3Faculty of Medicine, School of Health Sciences, University of Iceland, Laeknagardur 4th Floor, Vatnsmyrarvegur 16, 101 Reykjavik, Iceland; 4Department of Pediatrics, National University Hospital, Hringbraut, 101 Reykjavik, Iceland

**Keywords:** eating behaviors, fussy-eating, neurodevelopmental disorders, autism spectrum disorder, ADHD, problematic mealtime behaviors, parents-child dyads

## Abstract

Fussy-eating children often display problematic behaviors around mealtimes, such as irritation, opposition, or may even throw tantrums. This may lead to reduced food variety and poor nutritional profiles, which may increase parents’ worries about their children’s diet, particularly when the children also have neurodevelopmental disorders (ND) such as Autism Spectrum Disorder (ASD) and Attention-Deficit/Hyperactive Disorder (ADHD). To investigate the effect of Taste Education on problematic mealtime behaviors, 81 children aged 8–12 years, with ND (*n* = 33) and without (*n* = 48), and their parents, participated in a 7-week Taste Education intervention. Children were matched on age, sex, and ND, and allocated at random into Immediate-intervention and Delayed-intervention groups. Parents completed the Meals in Our Household Questionnaire (MiOH). To examine changes in MiOH-scores, repeated-measures analysis-of-variance with time-points were used, with condition as factors (Immediate intervention and Delayed intervention). Baseline measures were adjusted for, and a robust linear mixed-model was fitted. Results showed superior outcomes for Intervention compared to waiting on all measures of MiOH, with stable effects through six-month follow-up. Differences were non-significant between children with and without ND. The Taste Education program suggests a promising, simple, and non-intrusive way to reduce children’s problematic mealtime behaviors in the long term.

## 1. Introduction

Mealtime behavior problems frequently occur among families of children who display fussy or selective eating behaviors. There are many variations on the definition of problematic eating behaviors, including food neophobia (rejecting novel or unknown foods) [1,2], fussy or picky eating (the rejection of a large proportion of novel and familiar foods), being grumpy at mealtimes, and displaying oppositional behaviors or throwing tantrums [3,4,5]. Problematic eating behaviors may result in adverse health-related outcomes, and a low-variety diet in the long-term [1,6,7], including poorer nutrition profiles [8,9,10,11], obesity [12,13,14], and parental concern or stress [15,16,17].

The prevalence rates of problematic eating behaviors vary widely, ranging from 10–89% [18,19], and although young children between ages two and six most commonly show stringent food preferences and difficult eating behaviors [6,15,20], problematic eating behaviors are found in older children as well, and may affect both children and their families [21,22,23]. This is particularly true for children with ND (neurodevelopmental disorders), such as Autism Spectrum Disorder (ASD) which entails significant difficulties in communication and social interaction as well as repetitive and restricted interests and behavior, and Attention-Deficit/Hyperactivity Disorder (ADHD), which is characterized by attention deficits, hyperactivity and impulsive behavior [24,25,26,27,28,29,30].

### 1.1. Children’s Eating Behaviors and Parental Concerns

Due to children’s food refusals and sometimes limited diets, concerned and well-meaning parents (or other primary caregivers, hereafter referred to as parents), may increase pressure on the children to eat nutritious, but less-liked foods such as fruit and vegetables [25,31,32,33,34,35]. Pressure to eat, and other coercive methods sometimes used by parents, may create reciprocal stress, i.e., stress in children as they may refuse to eat or taste the foods offered, and parental stress in worrying about their children’s health [33,34,36]. The pressure may be even more pronounced when dealing with children with ND [28,37,38]. Children with ASD are more likely to display mealtime behavior problems compared to children who are typically developing (TD), and studies indicate that, in particular, problematic mealtime behaviors may be reactions to sensory overstimulation, as marked by rituals, inflexibility, repetitiveness, fear of novelty, and hypersensitivity to various stimuli, [39,40,41,42]. Research has also indicated that compared with TD children, children with ADHD may exhibit more disruptive mealtime behaviors, such as tantrums and noncompliance [5], poorer diets, such as fewer fruits and vegetables, and increased consumption of sugary foods and sweet drinks [43].

The strain, and sometimes burden of raising a child with special needs, may lead to numerous forms of challenging and disruptive communication across mealtimes, especially around children’s food choices and problematic mealtime routines [5,26,40]. These responses can unintentionally lead to decreased intake and acceptance of healthier foods [22,44,45]. For example, parents may manage stressful communications by making trade-offs between healthier foods and other family priorities, thereby lowering expectations for healthier foods [46,47].

### 1.2. Structure around Family Meals

Family mealtime routines, family meals, and food-related parenting practices are increasingly being appreciated as important aspects in the development of children’s eating habits [15,17,23,47,48]. Family dinners have been associated with a lowered risk for eating unhealthy foods [49,50], lower rates of obesity [51], better academic performance [52], and fewer behavioral problems in children [53], among other benefits. However, the health benefits of family meals may be mediated by more unified families, better communication during mealtimes [54], parenting styles [25,47], or children’s temperament [55]. Thus, creating a constructive or positive atmosphere at home may be challenging for families of children with ND [29]. In some instances, desired behaviors may be modified or rewarded with foods, e.g., when a child gets a dessert for sitting still at the dinner table, special treats for receiving good grades at school, or cookies for eating vegetables [56]. The use of food to reward good behavior or punish poor behavior is prevalent among parents, despite being a widely discouraged practice as it may unfavorably affect learning, behavior, and physical health [57]. However, when parents feel desperate or concerned about their child’s health or behavior, they may resort to various measures to improve the situation [58].

### 1.3. Communication around Food

Although well-meaning, parents are often unaware of the unhelpful methods they use to communicate to their children the necessities of eating healthily. In addition to children being rewarded for eating less desired foods such as fruit and vegetables, pressure to eat can manifest in several ways, even as praise, encouragement, and reassurance [25,28,37,57]. These methods are generally not helpful in increasing the consumption or variety of healthy foods in the long term [31,33,59]. In some studies, children’s problematic eating behaviors even increased after experiencing pressure, or unsolicited comments by parents and other family members [31,33], which in turn may increase children’s anxiety and, consequently, exasperate problematic eating behaviors [33,60]. The unhelpful and sometimes unkind words directed at children, the disapproval or disappointment over failures to eat the foods offered, or even fanfare when children finally do taste previously refused foods, are almost always a deterrent to children’s progress in tasting new foods. Subsequently, children, when under pressure, may then back out of trying new foods altogether [60,61,62,63]. Retrospective reports even suggest that adults’ food dislikes may be traced back to negative encounters concerning previous pressure to eat [31,64].

Negative communication around food and mealtimes may create adverse connotations and experiences, so much so that children may even refuse to come to the dinner table [21,27]. Negative experiences may even lead to family members eating separate meals, in separate rooms, at different times during the day [21,34,55,63]. Importantly though, parents can be valuable role models and may positively impact children’s eating and mealtime behaviors, especially when combined with family-based interventions [1,16,34].

### 1.4. Parental Involvement in Food-Based Interventions

Parental feeding behaviors play a critical role concerning young children’s food preferences and health [65,66] which is possibly why interventions for expanding the variety of fruit and vegetable in children, have increasingly involved parents, including parents of children with ND [65,67]. Likewise, the benefits of parental components in family-based treatments for children with obesity are well established [68,69]. There may be numerous benefits involved in cooking with children, such as more frequent family meals, healthier diets [1,70,71,72,73,74,75], and increased vegetable intake [72,73,76]. In a recent study, involving young children in cooking meals together was linked to parents’ lowered worries about fussy eating in their children [77]. Parents are especially valuable agents in influencing their children’s eating habits, so involving them in food-based study settings seems logical. Furthermore, since some parents lack the crucial abilities to mold children’s healthy eating habits, the parental element in food-based interventions is considered invaluable [67,72,73].

Researchers have increasingly called for new methods to deal with children’s problematic eating behaviors, including aiming for positive experiences around mealtimes [1,17,35,78]. Previously, food-based intervention studies with parental components, such as education on nutrition, and well-implemented reward-based systems, have provided promising results in dealing with some aspects of problematic eating behaviors, in both TD children and children with ND [16,67,72,73,79]. In one randomized controlled study on disruptive mealtime behaviors in young TD children, six months follow-up revealed support for the intervention’s utility [18]. Another study, on young children with ASD, comprised 11 sessions with an individually delivered parent training program where the outcomes were favorable but with small effect sizes [41]. Although sometimes effective, these interventions tend to be quite laborious and time-intensive, offered to younger children, and not inclusive of children with ADHD. Creating easy-to-use tools for parents to deal with their children’s difficult mealtime behaviors, is therefore of high importance as negative experiences and parents’ unwitting communication and responses may aggravate problematic eating behaviors in their children.

This study aimed to investigate the parenting-education component applied to parents in the Taste Education intervention, and whether the methods would improve problematic eating behaviors in children with and without ND.

## 2. Materials and Methods

### 2.1. Study Design and Randomization

The Taste Education study was a food-based intervention, a prospective, longitudinal, randomized controlled study compared with a matched, delayed intervention group. This current research is a part of a larger study on fussy eating and other problematic eating behaviors affecting families of fussy-eating children. Details of the study have been previously reported [80,81]. After an invitation to take part, families were matched on children’s age, sex, and ND-status and randomized either to an Immediate intervention group or a Delayed intervention group (Figure 1). A parallel data collection was performed to evaluate changes in problematic eating behaviors during the seven weeks when no intervention was conducted. The Delayed intervention groups were then provided the same program as the Immediate intervention groups, only delayed by seven weeks.

### 2.2. Study Setting

The intervention was performed within the School of Education’s home economics teaching kitchen, at the University of Iceland, from 2018 to 2019. This setting was specifically selected as a typical home-economics classroom familiar to Icelandic children. The Taste Education program was designed as after-school sessions delivered midweek, at the same time of day for all participants. Further, the parenting-education component was delivered in two seminars, separate from the children. Both parents (when applicable) were encouraged to participate in the sessions if possible.

### 2.3. Measures

Demographic information (children’s sex and age, as well as parental education, marital status, and occupation) was acquired from the screening questionnaires created by the researchers.

Meals in Our Household (MiOH) [21] is a parent-report questionnaire measuring six domains: (1) *Structure of family meals*, (2) *Problematic child mealtime behaviors* (frequency and magnitude), (3) *Use of food as reward*, (4) *Parental concern about child diet*, (5) *Spousal stress related to child’s mealtime behavior*, and (6) *Influence of child’s food preferences on what other family members eat*. The measure has been applied in research on 3–11-year-old TD children and children with ASD [82]. In other studies, the measure has shown good internal consistencies (Cronbach’s *α*) for the six domains [21]. For this current study, the internal reliability coefficients for all the domains ranged from good (*α* = 0.71) for *Structure of family meals*, to excellent for *Parental concern for child diet* (*α* = 0.90).

For domain one, *Structure of family meals*, parents described the frequency of 10 common family mealtime structures issues on a five-point Likert scale, where lower scores indicated less structure. Questions included “Meals in our household are rushed” and “Someone in our household cooks meals.” Four questions were reverse-coded. For domain two, *Problematic child mealtime behaviors*, parents rated on a four-point Likert scale the frequency with which their child displayed problematic mealtime behaviors, and if the behavior was problematic for the family (also on a four-point Likert scale). Higher scores revealed greater frequencies of problems and a larger magnitude of problems as estimated by the parents. Questions included: “My child refuses to come when it is time to eat,” and “How much of a problem is it that your child refuses to come when it is time?” For domain three, *Use of food as reward*, parents answered how often they used food to reward or encourage their child for certain behaviors. The responses were recorded on a five-point Likert scale, where higher scores indicated a higher frequency of using rewards. Finally, for domain four, *Parental concern about child diet*, parents were asked about concerns over what their child ate. Examples include “Child does not eat vegetables,” and “Child will not try new foods.” The responses were recorded on a six-point Likert scale where higher scores indicated greater worries about the child’s diet. One question was added by the researchers of the current study: “Child does not eat fish” as fish is a staple dish in the Icelandic diet [83]. Domains five and six, although interesting, were excluded as they were not the focus of the current study. Further, since the test battery administered within the Taste Education intervention was extensive [81], only the domains deemed essential were included in this current study.

Parents answered the MiOH at the beginning of the intervention. At the end of the seven weeks waiting period, questions were repeated for parents in the Delayed intervention group, then for all groups by the intervention end-point, including follow-up at six months (Figure 2). All questionnaires were processed and collected online using Qualtrics software (Qualtrics, Provo, UT, USA).

### 2.4. Recruitment

Parents of 8–12-year-old children who were all fussy eaters, with and without ND, were invited to participate in the intervention. Participants were invited via adverts on a website dedicated to the study, email lists in partnership with the Icelandic ASD and ADHD societies, and through adverts on social media.

### 2.5. Participants

Of the potentially 190 suitable parent-child dyads who finished the screening questionnaire for eligibility and consent, 95 (50.0%) agreed to take part, and 81 participants completed the intervention in full. Some of the prospective participants declined for various reasons (Figure 1). After the parents had filled out the questionnaire and consented to the study, they were matched by a child’s age, sex, and diagnosis status and were subsequently randomized into two groups. One child was withdrawn from the study after three sessions due to personal reasons. Five parent-child dyads dropped out after the parenting-education sessions due to their hectic schedules. Additionally, four children (three parents) withdrew from participating due to extraneous issues (for more details, see Thorsteinsdottir et al. [81]).

To guarantee validation of children’s ADHD and ASD diagnoses, all applicants were required to have been diagnosed by one of the three main Icelandic diagnostic centers, which all use standardized protocols and diagnostic instruments. All children attended mainstream schooling. Inclusion criteria encompassed fussy eating children 8–12 years of age and Icelandic-speaking parents. Data were pooled in both groups to increase statistical power for calculating overall changes in fussy eating from baseline to six months follow-up. Participants from the Immediate intervention (*n* = 38) and Delayed intervention group (*n* = 43) were allocated at random to either group, eventually receiving the same program. There were no significant differences between the Delayed and Immediate intervention groups for any parent or children’s background measures, including children’s age, sex, and diagnosis status (Table 1). The Delayed intervention groups were delayed by seven weeks and then received the same program as the Intervention groups.

### 2.6. Procedures

Parents volunteered to partake in the study and were invited to answer a screening questionnaire online about themselves and their children. Parents were advised that the intervention would not impede other services the child might be receiving somewhere else or exclude them from receiving them. Participants were not financially rewarded for their involvement in the study, although the children received a graduation certification and were gifted a tote bag for completing the Taste Education program.

### 2.7. The Taste Education Parenting-Education Sessions

The Taste Education sessions were divided into two components, i.e., (a) two parenting-education sessions, two hours each, prior to the first session with the children, and (b) six kitchen sessions, 90 min each, comprising food preparation skills and games as a base for sensory and taste education in the teaching kitchen with parent-child dyads (for further details of the kitchen sessions see Thorsteinsdottir et al. [81]). The parenting-education sessions were administered by a psychologist, and a nutritionist, who were also the two main taste educators. The sessions were delivered to parents (without the children) in groups one week before the first kitchen session. The group setting has been used in other food intervention studies [18,84].

The themes incorporated into the parenting-education sessions captured several topics, including basic behavior modification techniques (e.g., examples of negative and positive reinforcement), general nutrition education, fussy eating in children with and without ND, the disadvantages of using food as a reward, and the significance of parents’ duties in the Taste Education program and continued support and practice at home. The parents were encouraged to ask questions if they had any concerns about their children’s diet or food-related behaviors. They were also informed that within the study setting, a room would be dedicated to allowing the children to relax if they became overwhelmed with stimuli such as when interacting with other children, sounds, or smells. The parents were informed that the children were not required to taste anything if they did not want to and if they were anxious during the kitchen sessions, they were not obligated to participate. The parents were also asked not to take control in the kitchen and only to assist the children when needed (e.g., lifting heavy equipment) while still taking an active part in the sessions. They were also instructed to show enthusiasm towards anything the children participated in, without pressuring them into tasting anything. Parents were encouraged to taste food items when presented by a taste educator and neutrally or positively describe the experience, e.g., “I like the color on that grape” or “This smell reminds me of your grandma’s favorite food.” Parents were also encouraged not to show too much enthusiasm or excitement towards the children when they tasted something novel or unexpected, as this might create unwanted attention or pressure. They were also instructed not to comment in a derogatory way when the children were not interested in tasting the foods offered, e.g., “you never try anything you are offered,” or “why can’t you just taste this once?”

### 2.8. The Taste Education Sessions

The sensory-based education which was based on games, started with exercises on smells and textures in the first session, giving the children an opportunity to get used to the surroundings, noises, and smells of the kitchen area. The following five sessions introduced more texture-based exercises, sounds of food in the mouth, and cooking and baking simple recipes (re-introducing the now familiar ingredients). Each session was built upon the previous one with food items used previously, re-introduced in the following session. Exercises were based on repeated exposures (repetition of seeing foods, handling them, and tasting). Although acceptance of food items was not counted objectively in this current study, a broad variety of fruit, vegetables, legumes, and other less accepted foods by fussy eaters were laid out in every session. For details on the foods introduced in the Taste Education intervention, and changes in acceptance see Thorsteinsdottir et al. [81]. In two instances, over the six-week kitchen sessions, parents were given the opportunity to discuss the children’s progress briefly. The group setting, away from the children, allowed the parents to ask questions, receive feedback, and discuss any concerns they might have about having children with and without ND and problematic eating behaviors.

#### Instructions for Trainers

The Taste Education team was multidisciplinary, and leading each Taste Education session was a nutritionist and a psychologist. In addition, four assistant taste educators (a rotation of undergraduate and graduate students in psychology, nutrition science, pedagogy, home economics, and teaching) had a 2-h training seminar on the Taste Education methods and how to guide and assist children in the study gently. The taste educators received similar instructions to the parents regarding not pressuring the children into tasting any foods, and how to respond neutrally or positively to the foods used in the study. For example, if the child refused to taste food items, a taste educator was to offer the parent to taste and then move on to the next child. The taste educators observed parents in their communication with their children during all sessions. If a parent did not follow the guidelines discussed and encouraged in the parental education sessions, they were gently reminded of them by the taste educators who also modeled the appropriate response if needed. For example, if parents were tempted to pressure the child to eat a vegetable, the parents were reminded that the child did not have to eat it or even taste it, but instead might discuss the vegetable’s feature, smell, sound, and texture.

Following each session, the children were handed simple assignments to complete at home using a specially designed app. The assignments included a simple, themed task built on each session’s topic. The tasks were designed so children and parents could solve them easily, inciting discussions around the task and the foods used.

### 2.9. Statistical Analyses

When evaluating the intervention efficacy for the MiOH, completers only were used for the statistical analyses. Five participants did not complete all measurements but did have end-point measures and were contained in the linear mixed model evaluations between ND and non-ND groups. Data were analyzed using R version 4.1.3 (CoreTeam, R Foundation for Statistical Computing, Vienna Austria). [85]. A repeated-measures analysis of variance was used with time (pre-baseline, baseline, or post-intervention, and follow-up), and condition as factors (Immediate and Delayed intervention) were used to assess changes in the primary outcome (MiOH scores). The analysis adjusted for baseline measures.

ANCOVA-post for MiOH-scores following intervention in the Immediate intervention group were applied as the main test of efficacy. In particular, after adjusting for baseline scores, evaluating MiOH-scores at post-intervention for Immediate intervention and baseline for Delayed intervention were compared across groups (Figure 2). This approach is warranted as randomization permits us to safely assume that the population means at baseline were equal in both groups [86].

Data from the Immediate and Delayed intervention groups were pooled to evaluate differences in treatment between ND and non-ND children. Using dummy coding, a robust linear mixed model was fitted where MiOH-scores were regressed on time-point (baseline, post-intervention), ND-status (ND, non-ND), and their interaction. The interaction slope, therefore, becomes the difference in differences between baseline and post-intervention across the ND and non-ND groups. Wald confidence intervals for the effect of ND were provided as an indicator of the level of uncertainty in the estimate. However, the null hypothesis of no difference is not assessed, as ND is not a randomized factor, and it is, therefore, exceedingly implausible that the expected difference is precisely zero. The R package robustlmm 2.51 [87] was applied to fit the robust linear mixed models. Default tuning parameters were utilized.

## 3. Results

### 3.1. Sample Characteristics

Table 1 reveals the demographic characteristics of the 81 participants. No significant differences were found between any of the demographic or background measurements between either group: Immediate intervention or Delayed intervention. The majority of participants who completed the intervention were mothers (92.2%), although fathers were also involved and comprised, on average, 30% of the participants in the sessions. Around ten parent-child dyads were in each group, and approximately 95% of the children completed all sessions. All families spoke Icelandic, and the majority were living in the main capital of Reykjavík and surrounding towns. The children were 8–12-year-old (*M* = 10.4, *SD* = 1.43 at baseline), and 41.9% were female. More than half of the children had a diagnosed disorder (51.9%); 38.2% had ND, i.e., either ADHD, ASD, or both; 14.8% had Anxiety, and 22.2% had other disorders (e.g., developmental delays, mild cerebral palsy, mild learning difficulties, Oppositional Defiant Disorder (ODD) and Tourette syndrome). No children with ND were included in the TD group, and only children with ND (including Tourette syndrome) were included in the ND group.

### 3.2. MiOH Scores

After adjusting for baseline scores in the MiOH measurement (Figure 3), significant differences were identified in the *Structure of family meals* scores between Immediate intervention (post-intervention) and Delayed intervention (baseline), *F* (1, 83) = 6.410, *p* = 0.013. The adjusted mean score was significantly higher in the Immediate intervention group (*M* = 2.92, *SE* = 0.09) compared to the Delayed intervention group (*M* = 2.70, *SE* = 0.09), a mean difference of 0.22, 95% CI [0.05, 0.39]. Significant differences were also found in the *Problematic child mealtime behaviors* (frequency) scores between Immediate intervention (post-intervention) and Delayed intervention (baseline), *F* (1, 84) = 5.861, *p* = 0.018. The adjusted mean score was significantly lower in the Immediate intervention group (*M* = 1.56, *SE* = 0.09) compared to the Delayed intervention group (*M* = 1.73, *SE* = 0.09), a mean difference of −0.23, 95% CI [−0.41, −0.04]. Significant differences were detected in the *Problematic child mealtime behaviors* (magnitude), scores between Immediate intervention (post-intervention) and Delayed intervention (baseline) *F* (1, 84) = 5.325, *p* = 0.024. The adjusted mean score was significantly lower in the Immediate intervention group (*M* = 1.64, *SE* = 0.10) compared to the Delayed intervention (*M* = 1.43, *SE* = 0.10), a mean difference of −0.24, 95% CI [−0.44, −0.03]. When analyzing *Use of food as reward*, significant differences were detected between Immediate intervention (post-intervention) and Delayed intervention (baseline), *F* (1, 84) = 12.048, *p* < 0.001. The adjusted mean score was significantly lower in the Immediate intervention group (*M* = 0.67, *SE* = 0.10) compared with the Delayed intervention group (*M* = 0.95, *SE* = 0.10), a mean difference of −0.35, 95% CI [−0.54, −0.15]. When analyzing *Parental concern about child diet*, significant differences were identified between Immediate intervention (post-intervention) and Delayed intervention (baseline), *F* (1, 84) = 9.795, *p* = 0.002. The adjusted mean score was significantly lower in the Immediate intervention group (*M* = 1.55, *SE* = 0.09) compared with the Delayed intervention group (*M* = 1.96, *SE* = 0.09), a mean difference of −0.29, 95% CI [−0.48, −0.11] (Figure 3).

A robust linear mixed model yielded a point estimate of 0.19 for the interaction of ND-status and experimental group (95% Wald CI: 0.06–0.32) for *Structure of family meals*. For *Problematic child mealtime behaviors* (frequency), the point estimate for the interaction of ND-status and experimental group was 0.02 (95% Wald CI: −0.14–0.19), and for *Problematic child mealtime behaviors* (magnitude) −0.10 (95% Wald CI: −0.28–0.08). For *Use of food as reward* the point estimate was −0.20 (95% Wald CI: −0.46–0.05), and for *Parental concern about child diet* the point estimate was −0.01 for the interaction of ND-status and experimental group (95% Wald CI: −0.17–0.15) (Figure 4).

## 4. Discussion

The key objective of this study was to examine the parenting-education component applied to parents in the Taste Education food-based intervention, and whether these methods would effectively reduce problematic eating behaviors in children with and without ND, as measured with the MiOH questionnaire. Results showed superior outcomes for Intervention compared to waiting in all domains measured in this intervention: *Structure of family meals, Problematic child mealtime behaviors* (frequency and magnitude), *Use of food as reward*, and *Parental concern about child diet*. There were no significant differences in any domains between children with and without ND.

When comparing the Immediate and Delayed intervention groups on the MiOH, results suggested that the methods applied effectively decreased problematic mealtime behaviors, since the intervention demonstrated superior results compared to waiting. A continued reduction was identified from pre-baseline throughout the intervention for the Delayed intervention group on some of the domains, i.e., *Problematic child mealtime behaviors* (frequency) and *Parental concern about child diet*, with a similar drop in the Immediate intervention group for *Problematic child mealtime behaviors* (frequency and magnitude). In *Problematic child mealtime behaviors* (magnitude) *and Use of food as reward* domains, there was a fair rise between pre-baseline and baseline in the Delayed intervention group. Since the parents in the Delayed intervention had answered questions on the MiOH pre-baseline, they may have heightened their awareness and worries about their child’s behavior, which may have led to alterations in the MiOH scores. These changes were, however, not reflected in the other domains. At six months follow-up, the *Structure of family meals* scores were still improved and higher, and apart from *Use of food as reward*, the scores on the remaining domains were still improved and lower compared to baseline.

One of the core elements of the parenting-education of the Taste Education intervention was educating parents about how their responses may affect their children’s eating behaviors. For many parents, being able to ask questions about appropriate responses to problematic behaviors, and receiving feedback and validation from a healthcare professional, may be an essential part of confidently changing their children’s behavior [88,89], especially if the parents have concerns about their children’s diets or behaviors. In addition, parental behavior is a powerful influencing factor on children’s eating behavior, so reducing parental concern and stress may reduce stress in the children, which may play a part in problematic mealtime behaviors [90]. Further, spending time with the children in the Taste Education kitchen may also have been a protective factor, as studies have indicated that time spent with children in the kitchen is linked with lowered fussy eating [74] and may positively affect children’s eating behaviors [1,15,16]. Similarly, the social reinforcement deployed by the parents in our intervention [81] may also have facilitated the encouraging changes in the MiOH. This current study was unique in that a specific emphasis was placed on educating the parents about their children’s problematic eating behaviors, including fussy eating. In our study, considerable effort was put into teaching parents not to pressure their children into tasting the foods offered. The authors of the current study are not aware of any published studies where this emphasis on reducing the pressure on tasting is paramount. However, authors from a recent study did explicitly advise parents to relax their feeding efforts so as not to increase fussy eating and problematic eating behaviors [45]. Additionally, no published study specifically instructs parents not to show enthusiasm towards their children when tasting something novel or unexpected. This impulsive response from parents frequently happened during the early kitchen sessions when they got excited about the children’s progress. Again, this shows that well-meaning parents may not always know the appropriate responses to their children’s eating behaviors, although few studies recognize this.

There were no significant differences between children with and without ND regarding the interaction of ND-status and the experimental group at post-intervention. However, for *Structure of family meals*, *Problematic child mealtime behaviors* (magnitude), and *Parental concern about child diet*, children without ND seemed to fare a little worse in the follow-up than children with ND. This is interesting as it seems as though parents of children with ND may have utilized the methods in the intervention better than parents of children without ND and perhaps seen proportionally more change over time in their children, compared with children who did not have ND. Since children with ASD and ADHD are excluded from participation in many food-based studies, these results provide optimism in terms of an inclusive and simple method for changing problematic mealtime behaviors. This has been supported in our previous study where children with ASD, and children with ADHD were invited to participate in a food-based intervention alongside TD children, with similar results [80,81]. Problematic eating behaviors in children with ASD and/or ADHD may share certain commonalities, even more so than previously thought [91,92]. Furthermore, regardless of ND-status, most children with fussy eating share a fear of new foods, oppositional behavior, sensitivities to certain stimuli, anxiety, and dislike of fruit and vegetables and other fibrous or bitter foods [80,91,92,93]. Thus, as deployed in the Taste Education intervention, our gentle methods are perhaps suitable for children who share certain sensitivities regarding food and food-related stimuli.

### Strength and Limitations

The strength and limitations of the Taste Education intervention have been laid out previously [81], for example, the inclusive nature of the intervention, being the first of its type to invite children with ADHD in addition to ASD to participate. The study was also the first to include parent-child dyads investigating children with and without ND in a school setting. In addition, the study boasted very low drop-out rates, excellent attendance in sessions, a fairly high proportion of both parents attending, and very few non-completers [81]. The study also featured good to excellent internal consistency for the MiOH questionnaire, and randomization into Immediate and Delayed intervention groups.

This current study also had its limitations. Firstly, there was no comparison group with the parental-education session only, without the intervention [81]. The authors did initially consider this, but as the added benefits of involving parents are evident in other studies [16,18,67,79,84], and since almost half of the children had ND, and some of them needed additional support, we deemed it ethically important to run the intervention with the parents. Similar to our initial study [81], there was no evidence in this present study of children with ND performing worse than those without ND, although we do recognize that the results may not be replicated in lower functioning children with ASD, thus affecting generalizability and external validity. The children with ASD all attended mainstream schooling and were fairly high functioning, although some had repetitive behaviors, behavioral problems, ticks, and various sensory sensitivities. There are also no comparisons with other studies using the MiOH in Iceland, as ours was the first. However, the MiOH has been used in other studies with similarly aged children, TD children, and those with ND [21,22,94]. Another limitation is the parent-reported measure of the MiOH as parents reported on behavioral changes of their children, rather than objective examiners. Although a simple and low-cost way to obtain answers, parent-reported responses may be subject to bias and threats to validity and reliability. Finally, one of the greatest limitations of the Taste Education intervention was the low proportion of single-parent homes and a high proportion of educated parents in full-time employment. Using these measures as a proxy for socioeconomic status (SES), the results may not replicate in lower SES circumstances. Otherwise, the participants reflect the Icelandic population well. Participation was without charge, but the indirect cost may have included families taking time off from work or other commitments [81].

## 5. Conclusions

Overall, the results of this present study suggest that generally, participation in the Taste Education program shows promise as a short, non-invasive, simple, and effective way to improve mealtime behaviors, long-term, regardless of ND-status. Our results also indicate that the Taste Education methods may reduce parental concern about their children’s diet, which is of great value as parental concern may increase fussy eating and problematic eating behaviors. Our easy-to-apply methods may change how future food-based interventions are implemented, especially considering children with ND. These findings support the need for further studies since, although parental involvement has proven to be beneficial in other studies and is often seen as a key component, it is important to further test the parental-education component of the Taste Education program, possibly as a standalone feature, comparing seminars and educational materials. We believe the Taste Education program provides parents with the easy tools and gentle methods to improve children’s mealtime behaviors.

## Figures and Tables

**Figure 1 nutrients-14-04000-f001:**
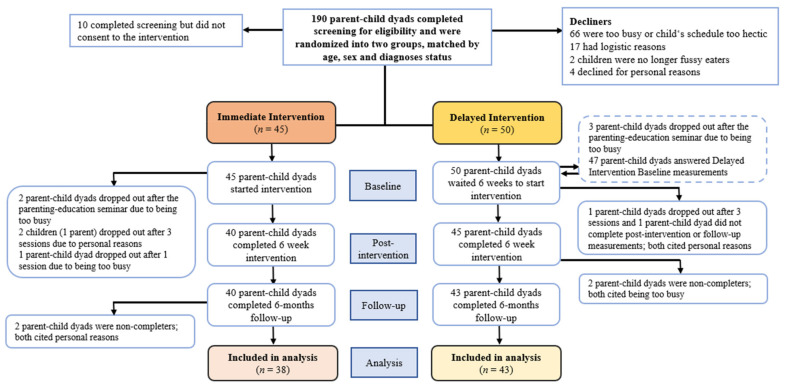
Flow diagram showing the intervention by stages of the study. In the Delayed intervention group, five siblings joined the children that had already been assigned to the group.

**Figure 2 nutrients-14-04000-f002:**
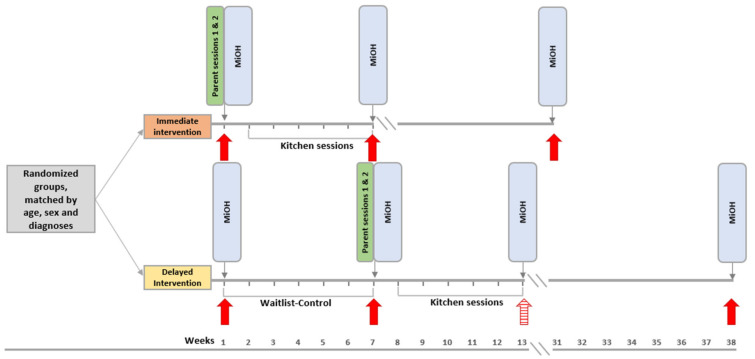
Study timeline showing time-points for pre-baseline and baseline assessments, post-intervention, and follow-up. *Note.* Abbreviation: MiOH, Meals in our Household Questionnaire.

**Figure 3 nutrients-14-04000-f003:**
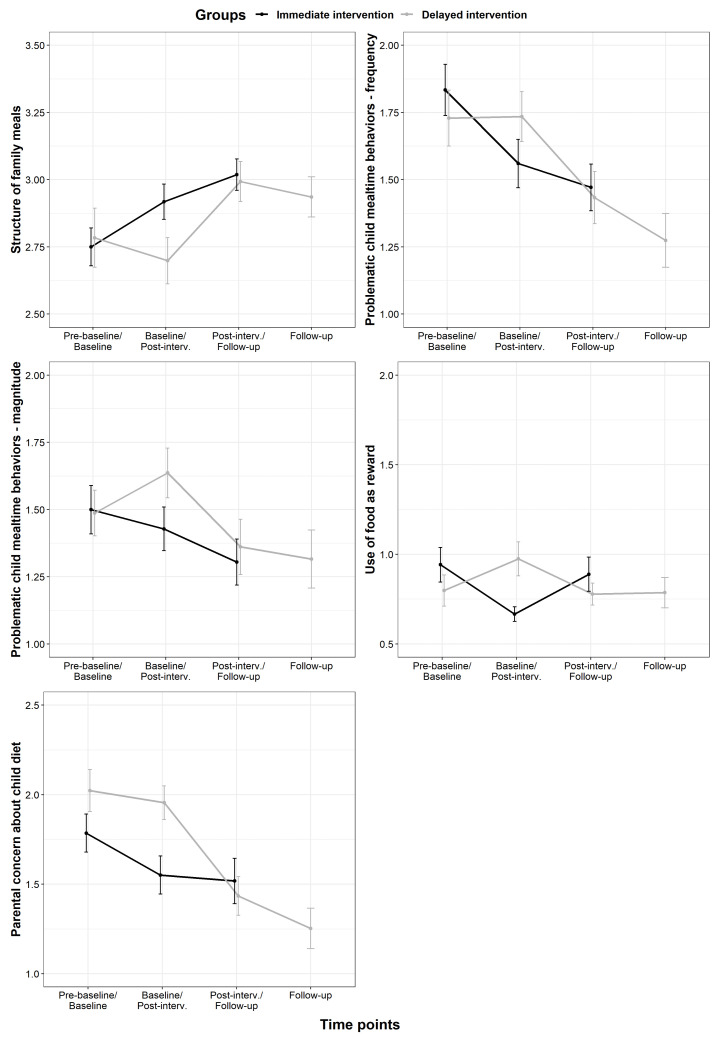
Mean changes with standard error of the means for Meals in Our Household (MiOH) scores for the domains between time points.

**Figure 4 nutrients-14-04000-f004:**
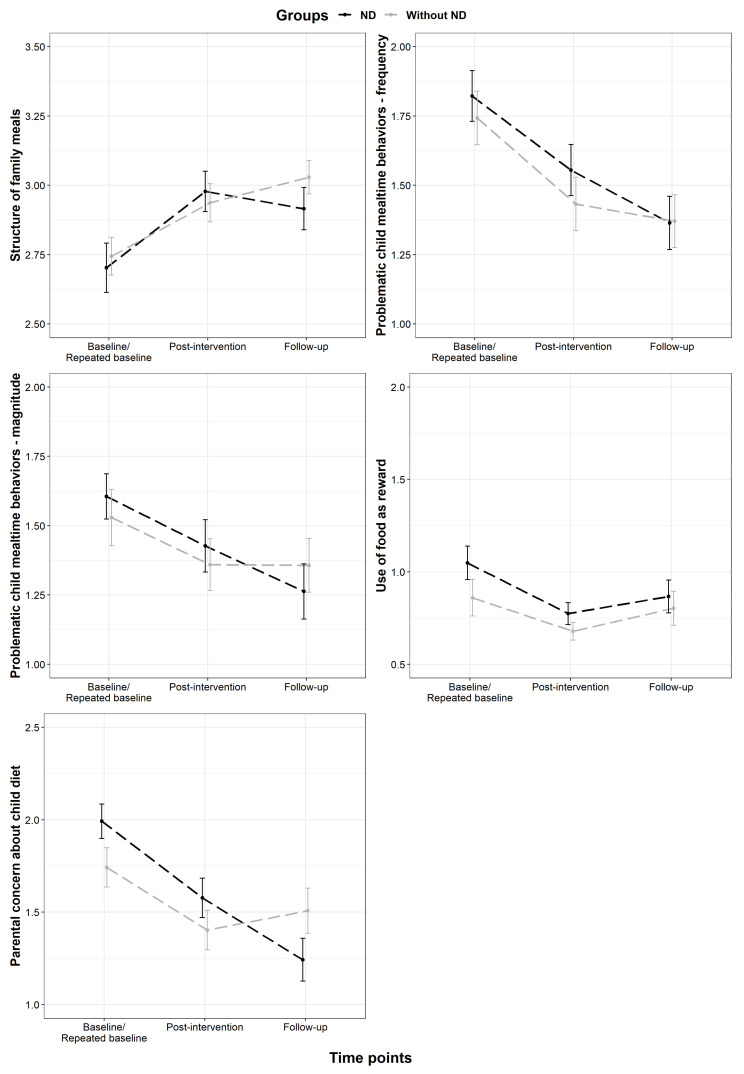
Mean changes with standard error of the means for Meals in Our Household (MiOH), for the domains, between time points, based on ND-status.

**Table 1 nutrients-14-04000-t001:** Characteristics of the total sample, background characteristics, and baseline demographic comparing participants in the Immediate and Delayed intervention groups.

	Overall *n* = 81	Immediate Intervention *n* = 38	Delayed Intervention *n* = 43
**Child**			
**Children’s age in years**, Mean (SD)	10.4 (1.43)	10.4 (1.23)	10.4 (1.47)
**Female**, *n* (%)	34 (41.9)	18 (47.4)	16 (37.2)
**Diagnoses**, *n* (%)			
ND (ADHD, Autism, or both)	31 (38.2)	16 (42.1)	15 (34.9)
ADHD, primarily	12 (14.8)	5 (13.2)	7 (16.3)
Autism, primarily	7 (8.6)	4 (10.5)	3 (7.0)
Anxiety	12 (14.8)	4 (10.5)	8 (18.6)
Other	18 (22.2)	11 (28.9)	10 (23.2)
No diagnoses	39 (48.1)	17 (44.7)	22 (51.2)
	**Overall** **^‡^*n* = 77**	**Immediate** **Intervention** ***n* = 37**	**Delayed** **Intervention** ***n* = 40**
**Parent**, *n* (%)			
Mother	71 (92.2)	35 (94.6)	36 (90.0)
**Education**, *n* (%)			
No higher education	3 (3.9)	1 (2.7)	2 (5.0)
Vocational education	13 (16.9)	8 (21.6)	5 (12.5)
University level	61 (79.2)	28 (75.7)	33 (82.5)
**Single parent household**, *n* (%)	10 (13.0)	6 (16.2)	4 (10.0)
**Occupational status**, *n* (%)			
Full-time occupation	58 (75.3)	29 (78.4)	29 (72.5)
Part-time occupation	10 (13.0)	4 (10.8)	6 (15.0)
Student	9 (11.7)	4 (10.8)	5 (12.5)
Other	8 (10.4)	2 (5.4)	6 (15.0)
**Children in household**, *n* (%)			
1	9 (11.7)	7 (18.9)	2 (5.0)
2	34 (44.1)	16 (43.2)	18 (45.0)
3	26 (33.8)	10 (27.0)	16 (40.0)
4 or more	8 (10.4)	4 (10.8)	4 (10.0)

*Note.* Abbreviation: SD, standard deviation. ^‡^ Five parents had two children (siblings) participating in the study.

## Data Availability

Data is available: The datasets generated during and/or analyzed during the current study are available from the corresponding author upon reasonable request.

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
