# Peer review of "Changes in Eating Behaviors Following Taste Education Intervention: Focusing on Children with and without Neurodevelopmental Disorders and Their Families: A Randomized Controlled Trial"

_nutrients, 2022, doi:10.3390/nu14194000_

Round 1

Reviewer 1 Report

This is a clearly-written manuscript describing results from a randomized controlled trial of a taste education intervention to improve mealtime behaviors in children with and without neurodevelopmental disorders. The authors address an important, yet understudied, problem concerning family mealtime difficulties, which could lead to low child diet quality. The intervention concept is interesting, and the analytic approach seems appropriate. The strengths of the design include the randomized treatment order, the repeated measures, and the inclusion of at least two family members. The limitations of the study design include the very small sample size (n=81 children), the use of a self-report (rather than objectively measured) outcome measure, and the use of a convenience sample with potentially limited generalizability.

Main weaknesses of the manuscript include a lack of sufficient detail describing the intervention components, and the descriptions of the strengths and limitations are not properly done. Additional detail regarding the messages of the intervention that were delivered to parents and children would be helpful as well as details regarding which foods children were prompted to try (and maybe even some data on how many children tried them versus refused). In terms of the strengths and limitations, the authors should revise this section to be focused solely on the strengths and limitations of the study design insofar as how they impact internal and external validity. As currently written, this section is primarily focused on the novelties or the study, which are fine to note, except they are not relevant to internal/external validity. I also recommend that the results from the main analyses comparing the Immediate Intervention post-intervention to the Delayed intervention baseline MiOH scores be presented in a table rather than written out in text (i.e., section 3.2).

Author Response

  1. I found the paper well-written and was excited to find an intervention that works for TD and ND children alike. While the rationale of working with parents on their feeding behavior is well argued, the intervention is not described in any detail - what were the activities in the school kitchen? This is a weakness of the paper, the Methods section should include a detailed descrption of the activities, or if the description is too long it should be presented as supplemental material.

Thank you for reviewing our paper and providing us with feedback. As per your suggestion, the description of the intervention has been elaborated on, but in order to keep the article fairly concise while adhering to the journal’s page limits, this section could unfortunately not be expanded to include a thorough description.  

  1. My main concern is that this same group published another paper on this particular study showing excellent results on the CEBQ published in Appetite last year, reference 81 in the current manuscript. Again as in the current paper, (which shows results for the MiOH questionnaire) the TD and ND children improved on all the CEBQ subscales. So there is already a published paper showing the exact same sample and intervention, with similar results. Moreover, the authors refer to this paper suggesting that the methods are the  better described there, but they are not. Just as in the current manuscript we learn of the frequency and number of meetings, the setting and time frame but not of the actual content of the intervention meetings. This makes the intervention "magical" and the study not replicable.

We understand this comment and the reviewer’s concern. To resolve this issue, we have added details to the description of the content of our intervention. We would like to emphasize that for our Appetite article we only used two of the CEBQ subscales where one (Food fussiness) showed positive results, and the other (Enjoyment of eating), although in the right direction, did not provide significant differences between intervention and waiting. For our current paper, we used a different measure, i.e., the MiOH questionnaire. Therefore, domains such as Use of Food as Reward, Structure of Family Meals, Parental Concerns about Child Diet, and Problematic Mealtime Behaviors were not covered in our Appetite article. Since the Appetite article did not cover these domains, we were very curious to see the results for MiOH since our parenting-education component covered these domains.

  1. I am not sure that showing the same results using a different questionnaire merits another peer-reviewed publication, and in any case, if to avoid self-plagiarism,  it has to be made clear that this is another paper on the same RCT.

Thank you for your comment. As identified above in response to comment #2 the current paper presents original results from MiOH which covers problematic eating behaviors beyond fussy eating. We have also ensured that the novelty of this current paper is stated in the methods section (page 4, section 2.1, line 152), i.e., that this paper is a part of a larger study that has been published elsewhere, to make the reader aware of the previous publication. We recognize that the wording for the methods section especially, may be similar to our methods section in our Appetite article and we have reviewed the wording and made sure that they are not the same to avoid self-plagiarism.

  1. A minor point - there was a typo on line 462 it should be "fare" and not "fair"

Thank you, the typo has been corrected (page 14, section 4, line 496).

Reviewer 2 Report

I found the paper well-written and was excited to find an intervention that works for TD and ND children alike. While the rationale of working with parents on their feeding behavior is well argued, the intervention is not described in any detail - what were the activities in the school kitchen? This is a weakness of the paper, the Methods section should include a detailed descrption of the activities, or if the description is too long it should be presented as supplemental material.

My main concern is that this same group published another paper on this particular study showing excellent results on the CEBQ published in Appetite last year, reference 81 in the current manuscript. Again as in the current paper, (which shows results for the MiOH questionnaire) the TD and ND children improved on all the CEBQ subscales. So there is already a published paper showing the exact same sample and intervention, with similar results. Moreover, the authors refer to this paper suggesting that the methods are the  better described there, but they are not. Just as in the current manuscript we learn of the frequency and number of meetings, the setting and time frame but not of the actual content of the intervention meetings. This makes the intervention "magical" and the study not replicable.

I am not sure that showing the same results using a different questionnaire merits another peer-reviewed publication, and in any case, if to avoid self-plagiarism,  it has to be made clear that this is another paper on the same RCT.

A minor point - there was a typo on line 462 it should be "fare" and not "fair"

Author Response

  1. Thank you for the opportunity to review the article “Changes in eating behaviors following Taste Education intervention: Focusing on children with and without neurodevelopmental disorders and their families. A randomized controlled trial”. This study whose aim was to develop a randomised control trial to study the parenting-education component applied to parents in the Taste Education intervention, and its effectiveness in reducing problematic eating behaviors in children with and without neurodevelopmental disorders (ND), is a well conducted study. It is clearly written and contributes to the literatura of eating behaviours in children with ADHD and ASD. A few changes could be made to improve the paper.

Thank you for reviewing our paper, we are most grateful for your insight.

  1. Figure 2. Please include a footnote of the abbreviation fro MiOH

Resolved. A footnote has been added below Figure 2 (page 6, section 2.3, line 214).

  1. In table 1, could you include a column with the results for the differences between the immediate and delayed intervention groups.

Thank you. There is a sentence (page 6, section 2.5, line 240) stating that there were no significant differences in any of the demographic or background measures presented in Table 1.

  1. Figure 4 appears to be missing.

Most peculiar! It was actually Figure 3 that was missing, and Figure 4 was in its place. Our apologies, we have rectified this (page 11, section 3.2, line 428; page 13, section 3.2, line 440).

Reviewer 3 Report

Thank you for the opportunity to review the article “Changes in eating behaviors following Taste Education intervention: Focusing on children with and without neurodevelopmental disorders and their families. A randomized controlled trial”.

This study whose aim was to develop a randomised control trial to study the parenting-education component applied to parents in the Taste Education intervention, and its effectiveness in reducing problematic eating behaviors in children with and without neurodevelopmental disorders (ND), is a well conducted study. It is clearly written and contributes to the literatura of eating behaviours in children with ADHD and ASD. A few changes could be made to improve the paper.

Figure 2. Please include a footnote of the abbreviation fro MiOH

In table 1, could you include a column with the results for the differences between the immediate and delayed intervention groups.

Figure 4 appears to be missing.

Author Response

  1. This is a clearly-written manuscript describing results from a randomizedcontrolled trial of a taste education intervention to improve mealtime behaviors in children with and without neurodevelopmental disorders. The authors address an important, yet understudied, problem concerning family mealtime difficulties, which could lead to low child diet quality. The intervention concept is interesting, and the analytic approach seems appropriate. The strengths of the design include the randomized treatment order, the repeated measures, and the inclusion of at least two family members. The limitations of the study design include the very small sample size (n=81 children), the use of a self-report (rather than objectively measured) outcome measure, and the use of a convenience sample with potentially limited generalizability.

Thank you for your comments and feedback for which we are grateful. We have noted most of the limitations within the Strength and limitations section (page 15, section 4.1, lines 517-548) and added a sentence regarding the limitations of self-report measures.

  1. Main weaknesses of the manuscript include a lack of sufficient detail describing the intervention components, and the descriptions of the strengths and limitations are not properly done. Additional detail regarding the messages of the intervention that were delivered to parents and children would be helpful as well as details regarding which foods children were prompted to try (and maybe even some data on how many children tried them versus refused). In terms of the strengths and limitations, the authors should revise this section to be focused solely on the strengths and limitations of the study design insofar as how they impact internal and external validity. As currently written, this section is primarily focused on the novelties or the study, which are fine to note, except they are not relevant to internal/external validity. I also recommend that the results from the main analyses comparing the Immediate Intervention post-intervention to the Delayed intervention baseline MiOH scores be presented in a table rather than written out in text (i.e., section 3.2).

Thank you for this helpful comment. We have added more details to the intervention description as per your suggestions (sections 2.8 and 2.8.1). We have also revised and refocused the strength and limitation section and moved the paragraph on novelty into the Discussion section. We hope that both sections are now clearer and stronger.

Regarding foods that the children tried, and acceptance rates, although not within the scope of this current study as we were more concerned with behavioral problems related to food refusals, we reported on those results in our Appetite article (detailing the foods used in the study), although not counting foods or weighing them. We identified this as a limitation within said article.  

Note: Reviewer 2 pointed out that Figure 3 was most likely deleted somewhere along the way, and in fact Figure 3 was missing and Figure 4 was (incorrectly) in its place. We have rectified this and hope that the results within section 3.2 are now clearer. If a table would still make the results for section 3.2. clearer, we would be happy to add it.

Round 2

Reviewer 2 Report

This is an interesting and novel study and the manuscript is well written. The revision gives a much better description of the intervention and is thus more convincing. I found it very interesting, and was happy to have the opportunity to review.

Reviewer 3 Report

The changes that have been made to the paper are sufficient to warrant the acceptance of the paper.